# Efficacy of Copper Ion Treatment on Bacteria and Antibiotic Residues Contained in Bovine Waste Milk

**DOI:** 10.3390/antibiotics13111085

**Published:** 2024-11-14

**Authors:** Fernando Ulloa, Martina Penati, Constanza Naegel, Carlos Tejeda, Miguel Hernández-Agudelo, Pamela Steuer, Miguel Salgado

**Affiliations:** 1Escuela de Graduados, Facultad de Ciencias Veterinarias, Universidad Austral de Chile, Valdivia 5090000, Chile; fernando.ulloa@uach.cl (F.U.); jmiguel.hernandez@udea.edu.co (M.H.-A.); 2Instituto de Medicina Preventiva Veterinaria, Facultad de Ciencias Veterinarias, Universidad Austral de Chile, Valdivia 5090000, Chile; constanza.naegel@alumnos.uach.cl (C.N.); carlostb81@gmail.com (C.T.); pamelasteuer@gmail.com (P.S.); 3Department of Veterinary Medicine and Animal Science—DIVAS, University of Milan, 26900 Lodi, Italy; martina.penati@unimi.it

**Keywords:** waste milk, copper ion treatment, microorganisms, antibiotics

## Abstract

**Background/Objectives**: Waste milk harbors many bacteria and antibiotic residues. Calves fed with untreated waste milk have a higher incidence of scours and an increased risk of developing antimicrobial-resistant bacteria. This study aimed to evaluate the efficacy of treatment with copper ions on bacteria and antibiotics contained in bovine waste milk. **Methods**: Waste milk samples were collected from a dairy farm for seven weeks and were subjected to treatment with copper ions. Total bacterial counts, coliforms, *Streptococcus*, and *Staphylococcus* were assessed before and after treatment. Additionally, metagenomic analysis was performed to determine microbial diversity. **Results**: Before treatment, the total bacterial count average was 4.0 × 10^6^ CFU/mL, 1.7 × 10^4^ CFU/mL for coliforms, 2.6 × 10^6^ CFU/mL for *Streptococcus*, and 5.4 × 10^2^ CFU/mL for *Staphylococcus* Copper treatment significantly reduced bacterial counts within 15 min. Total bacteria decreased from 4.0 × 10^6^ CFU/mL to 1.1 × 10^2^ CFU/mL after 30 min; meanwhile, other groups were not detected. The most abundant groups were *Lactococcus* (29.94%), *Pseudomonas* (28.89%), and *Enterobacteriaceae* (21.19%). Beta-lactams were detected in five-sevenths samples, and in one sample they were detected before and at 15 min of treatment but not after 30 min. **Conclusions**: The effect of treatment with copper ions on the different bacterial groups was significantly effective but showed limited effect on the detection of antibiotics.

## 1. Introduction

Raw bovine milk contains diverse populations of microorganisms that can come from the teat canal, the surface of the udder, feces, cow bedding, air, water, soil, milking equipment, and milkers’ hands [1]. When milk is processed for human consumption, most of these microorganisms are eliminated through thermal processes, transforming the milk into a safe product [2]. However, milk from cows with intramammary infections treated with antibiotics has been called waste milk (WM), and it is mandatory to eliminate it and not market it. This also applies to milk contaminated with other types of drugs, mastitic milk with flocs of pus, fibrin strands, mammary effusion, etc., milk with high somatic cell counts (SCC), and post-colostral transition milk [3].

Waste milk is still widely used by dairy producers for feeding calves [4,5]. Despite the fact that its use is controversial due to its role in the transmission of pathogens [6] and the risk it poses of the emergence of bacteria resistant to antibiotics [3]. For instance, a study performed in the USA revealed that a significant proportion (82%) of WM used for feeding calves contained residues of beta-lactam antibiotics [7]. The presence of these residues is known to contribute to the selection of antibiotic-resistant bacteria [8]. Feeding calves with waste milk containing antibiotic residues appears to play an important role in the appearance of resistant bacteria in calves [9,10]. Nevertheless, it seems this is a more complex phenomenon that includes other factors such as age and intestinal microbiome development [9].

Some studies indicate that feeding calves untreated WM can negatively impact their health. Calves that have been fed untreated WM experience higher rates of diarrhea and lower weight gain compared to those fed with bulk tank milk. Additionally, it disrupts their gut microbiome, leading to a decline in bacterial alfa diversity and beneficial bacteria [10].

Although the use of thermal processes, such as pasteurization, has been considered effective in controlling viable microorganisms in raw milk [11], evidence has been shown that some pathogens, such as *Mycobacterium avium* subsp. *paratuberculosis* (MAP), have an ability to resist this thermal process [12]. In addition, doubts have also been raised about the high cost of pasteurizing equipment and the time and training required for its operation. All of these considerations have highlighted the need to explore other measures to avoid the transmission of infections through WM [13].

Various methods for degrading antibiotic residues in this milk have been evaluated, such as incubation with beta-lactamases, degradation through fermentation with beta-lactamase-producing bacteria, degradation with a combination of ultrafiltration and electrochemical oxidation, and degradation through an increase in heat or pH. However, it has not been possible to employ any of these methods in the field [3].

Heavy metals such as silver, zinc, and copper have traditionally been used in medicine for their antibacterial effects [14]. Indeed, copper is an essential micronutrient for most microorganisms; however, it is toxic at high concentrations [15]. Numerous studies have demonstrated the antimicrobial properties of copper in eliminating important animal and human pathogens, such as methicillin-resistant *Staphylococcus aureus* (MRSA) [16], *Enterobacter aerogenes*, *Pseudomonas aeruginosa*, *Escherichia coli* O157:H7, and *Mycobacterium bovis*, along with viruses and fungi [14].

In previous studies, it was demonstrated that a treatment based on copper ions was able to significantly reduce the viability and load of MAP both in buffer [17] and in milk naturally contaminated with this pathogen [13]. Likewise, it has been shown that treatment with copper ions on penicillin-contaminated milk and phosphate-buffered saline samples potentially affects antibiotic detection [18]. Considering this, the present study aimed to evaluate the efficacy of copper ion treatment in reducing pathogenic bacteria and affecting antibiotic residues in WM.

## 2. Results

### 2.1. Microbiological Analysis

The total bacterial count in the samples before the treatment with copper ions showed an average of 4.0 × 10^6^ CFU/mL, with a range of 3.9 × 10^4^–2.4 × 10^7^ CFU/mL. Coliforms were detected in all samples with an average of 1.7 × 10^4^ CFU/mL and a range of 5.2 × 10^2^–4.5 × 10^4^ CFU/mL. There were very high counts of *Streptococcus* spp. in all samples, exceeding 3.0 × 10^6^ CFU/mL, except in sample number 5, where the count was 1.3 × 10^4^ CFU/mL. As for *Staphylococcus*, it was detected in all samples at variable concentrations, with an average of 5.4 × 10^2^ CFU/mL and a range of 1.2 × 10^2^–1.2 × 10^3^ CFU/mL (Table 1).

Regarding specific pathogens, only *Listeria* spp. were detected in sample 1, while neither *Prototheca*, *Salmonella* spp., nor *Mycoplasma* was detected using the special cultures, nor was MAP or *M. bovis* detected by qPCR. Additionally, neither ESBL *E. coli* nor MRSA was identified in the suspected samples.

### 2.2. Metagenomic Analysis

In the metagenomic analysis of sample 4, the species, genera or families with an abundance greater than 1% were *Pseudomonas*, *Lactococcus raffinolactis*, *Lactococcus lactis*, *Escherichia coli*, *Lactococcus* spp., *Enterobacteriaceae*, *Pseudomonas* spp. *Kluyvera*, *Streptococcus uberis*, *Chryseobacterium*, *Acinetobacter,* and *Serratia* (Table 2).

Additionally, when grouping these taxa into genera or families, the most abundant groups obtained were *Lactococcus* (29.94%), *Pseudomonas* (28.89%), *Enterobacteriaceae* (21.19%), *Streptococcus* spp. (6.29%), *Chryseobacterium* (3.21%), *Acinetobacter* (2.69%), and *Serratia* (1.74%), which were also identified through metabarcoding of the *16s RNA* gene.

Among the taxa identified in the metagenomic analysis were some groups of bacteria reported at a low percentage of abundance (less than 1%), but which are important pathogenic agents for animals or humans, such as *Coxiella burnetii*, *Listeria* spp., *S. aureus*, *Histophillus somni*, *Klebsiella* spp., *Pseudomonas aeruginosa*, *Corynebacterium* spp., *Fusobacterium necrophorum*, *Rhodococcus* spp., and *Trueperella* spp.

### 2.3. Effect of Copper Ion Treatment on the Bacterial Population in Waste Milk Samples

The overall results of this study showed that copper ion treatment resulted in a significant reduction (*p* < 0.05) of TBC, *Streptococcus* spp., *Staphylococcus* spp., and Coliforms. The TBC decreased from 4.0 × 10^6^ CFU/mL to 2.7 × 10^3^ CFU/mL after 15 min and 1.1 × 10^2^ CFU/mL after 30 min. Furthermore, after 30 min. of treatment, TBC was only recorded in samples 1, 2, 4, and 5. *Streptococcus* were counted after 15 min of treatment in samples 1 and 2, with an average of 1.5 × 10^2^ CFU/mL after 15 min and no growth after 30 min of treatment. Meanwhile, *Staphylococcus* spp. decreased to 1.1 × 10^2^ CFU/mL after 15 min, and there was no growth after 30 min. Finally, coliforms had an average of 1.8 × 10^2^ CFU/mL after 15 min of treatment, and no growth was recorded after 30 min (Table 3). The percentage of bacterial reduction at 30 min was 100% for *Streptococcus*, *Staphylococcus*, and coliforms, and >99.99% for TBC.

The results of the *Listeria* culture showed growth in only one sample before treatment, with no detection after 15 or 30 min of treatment. Regarding *Salmonella* spp., *Mycoplasma* spp., and *Prototheca*, no growth was observed in the samples before treatment and after 15 or 30 min. of treatment with copper ion.

### 2.4. Copper Quantification

Regarding the copper concentrations before and after treatment, the average copper concentration of the WM samples was 0.06 mg/L (CI 0.03–0.08) before treatment, 262.08 mg/L (CI 247.32–276.84) after 15 min of treatment, and 650.32 mg/L (CI 628.07–672.58) after 30 min of treatment.

### 2.5. Effect of Copper Ion Treatment on the Detection of Antibiotics

Residues of beta-lactam antibiotics were found in samples 1, 2, 3, 5, and 7. No antibiotics were detected in samples 4 and 6. Regarding the detection of antibiotics in the samples after treatment with copper ions, there were no observed changes in antibiotic detection after 15 min of treatment. However, after 30 min of treatment, no antibiotics were detected in sample 5 (Table 4). The detection test on sample 7 for 15 and 30 min of treatment was invalidated, in accordance with the manufacturer’s reading instructions.

## 3. Discussion

Due to the very well-known antibacterial capacity of copper [16], it made sense to evaluate the effectiveness of copper ions under field conditions to broaden the scope of the findings reported. This led to the development of a treatment protocol that allows the release of copper ions, which are capable of inflicting oxidative damage not only to microorganisms in this milk but also to antibiotic residues in it.

As expected, a diverse range of bacteria was identified in the WM samples, including some known causative agents of intramammary infections in cows [19]. Notably, the TBC consistently indicated a high bacterial load in the WM, suggesting the presence of a significant population of both environmental and pathogenic bacteria [20,21]. The high counts of *Streptococcus* (mostly aesculin-positive) likely reflect environmental contamination, as these bacteria are commonly found in the surrounding environment [22]. Similarly, the high counts of coliform bacteria suggest both environmental and fecal contamination of the WM [21]. This finding could be indicative of inadequate hygiene practices during the handling of milk or feeding equipment [23,24]. High loads of all these bacterial groups, especially coliform bacteria, can result in infection in calves [25]. Proper milk storage and the cleaning of feeding equipment are important to prevent the carryover of diarrhea-causing pathogens [24,25].

We also investigated the presence of *M. bovis* and MAP in the WM samples, but no evidence of either pathogen was detected. In the case of *M. bovis*, this finding aligns with the herd’s official bovine tuberculosis-free status, established by the national control program. For MAP, the negative results could be attributed to several factors. First, the sampled cows might not have been high shedders of the bacteria [26], and second, a dilution effect could have occurred, reducing the bacterial concentration below the detection limit. It is important to acknowledge that WM is not the optimal sample type for detecting MAP or *M. bovis* infections. While milk samples can be used for such purposes [27], fecal or blood samples generally offer greater specificity and sensitivity for these pathogens.

Due to the limitations of culture-based methods, which only detect bacteria that can grow under specific conditions, an auxiliary test was employed to complement the information obtained from the culture [1]. By employing metagenomics, a more sensitive tool was obtained for identifying microorganisms in WM. This technique offers the advantage of identifying bacteria that are either non-culturable or require specialized laboratory conditions for growth. The metagenomic analysis revealed a highly diverse bacterial community, with some abundant groups coinciding with those detected through culture, such as *Pseudomonas* and *Streptococcus*-like organisms, including *Lactococcus*. *Pseudomonas* and *Lactococcus* species are commonly found in raw cow’s milk and play significant roles in dairy processing. While *Lactococcus* species are essential starters in cheesemaking, *Pseudomonas* spp. is often associated with milk spoilage, particularly in cold-stored milk due to its psychrotolerant nature [1].

Metagenomic analysis also revealed the presence of *S. aureus*, a significant pathogen responsible for intramammary infections in dairy cows. This pathogen is one of the most prevalent mastitis pathogens in Chile [28], making this finding unsurprising given the herd’s history of mastitis. Studies have shown that intramammary infections can occur in heifers before their first calving [29,30,31]. This finding raises concerns that feeding female calves with contaminated WM containing *S. aureus* may increase their risk of becoming colonized with this pathogen [32]. Therefore, it is highly advisable to not feed females with untreated WM or not use WM from cows with *S. aureus* intramammary infections.

The metagenomic analysis identified the genomes of bacteria that affect human and animal health, such as *Coxiella burnetii*. Regarding this pathogen, Hernández-Agudelo et al. (2023) have reported evidence of its presence in 76% of bulk tank milk samples from dairies in Southern Chile, which may suggest that this microorganism is widely distributed in the dairy environment, but only a few cases of sick animals or humans have been described [33].

Treatment with copper ions on bovine WM in this study showed significant efficacy, reducing the concentration of total bacteria by more than 99.9% after 15 min of treatment with less than 150 CFU/mL detected after 30 min of treatment. These results are not surprising since Steuer et al. (2021) obtained similar results in the reduction of bacteria and evaluated, under an experimental design, the effectiveness of treatment with copper ions on microorganisms present in bulk tank milk [13]. The survival of some microorganisms after 30 min of treatment is likely explained more by their extreme abundance than by the lower effectiveness of the treatment since the decrease in load after 30 min was still significant. Among the factors that influence the effectiveness of the treatment, the number of bacteria present in the sample, the number of cows with intramammary infections included in the sampling, and the possible resistance of some bacteria to copper should be considered [13]. When comparing the effectiveness of treatment with copper ions to that of pasteurization or acidification, we see that the latter does not lead to the complete elimination of bacterial groups before 30 min [20]. Additionally, for the copper treatment, the variables time and amperage can be controlled to achieve higher effectivity in less time of treatment.

In this study, the presence of antibiotic residues could be detected in 70% of the WM samples. The antibiotics used in the herd during sampling were cefquinome, oxytetracycline, cephapirin, and penicillin G procaine combined with dihydrostreptomycin. The residues were mainly beta-lactams. However, antibiotics from the tetracycline group were not detected in the analyzed samples. The study by Pereira et al. (2014) describes various antimicrobials that can be found in WM, such as ceftiofur, penicillin, ampicillin, and oxytetracycline [7]. In Chile, there are no studies on the use of antibiotics in dairy production. However, the majority of antibiotics registered for use in dairy cows belong to the group of beta-lactams and tetracyclines [7].

Regarding the effect of copper on the antibiotic groups, the result shows evidence that copper ions have an effect on the detection of antibiotic residues in the WM after a 30 min treatment, but this result is not conclusive because it was only evident in one of the samples (sample 5). This could be because the sample had a low concentration of antimicrobials or there were interactions with different types of antibiotics or the fat content [34]. The time of treatment, the concentration of antibiotics, and the percentage of fat are described as the main factors that influence the degradation of antibiotics as tetracyclines [34,35]. There are studies in which the elimination of antibiotics through an electrochemical process is evident [35]. Because quantitative methods were not used for the determination of antibiotics, it is not possible to associate the efficacy of treatment with copper ions with the concentration of antibiotics in the sample.

While copper ion treatment has shown promising results in eliminating bacteria from WM, a comprehensive evaluation of its application in calf feeding and its potential impact on calf health is essential. Ongoing monitoring of calf health parameters, such as growth rate, feed intake, and nutrient utilization, is crucial to assess the overall safety and efficacy of this treatment strategy. Furthermore, to ensure the innocuity of the treated WM for calf consumption, it is necessary to evaluate the efficacy of the copper removal process after treatment. Various methods, including membrane filtration, reverse osmosis, ion exchange, electrochemistry, chemical precipitation, adsorption, and biotechnology, can effectively remove copper ions [36]. However, it is crucial to consider the potential risks associated with copper-based treatments. Copper can exert selective pressure on bacteria, leading to the emergence of resistant strains. Co-selection mechanisms such as horizontal gene transfer can facilitate the dissemination of resistance genes in the environment [37]. While the copper ion treatment in this study effectively reduced bacterial populations, further research is necessary to assess the potential for surviving bacteria to develop increased resistance to copper and antibiotics.

## 4. Materials and Methods

### 4.1. Farm Description

This study involved WM collected from a dairy located in the region of Los Lagos, Chile. The herd is made up of about 1400 lactating animals, predominantly Jersey and Kiwi crossbreeds. The animals are managed under a seasonal system with permanent rotational grazing of a combination of permanent pastures, annual and biennial pastures, and supplementary crops. The average milk production per lactation stands at 4500 L. The cows are milked twice daily in a rotary-type parlor. The farm has a 5000 L bulk milk tank to collect all the WM produced.

### 4.2. Sampling of Waste Milk

Seven WM samples were collected from cows with mastitis, those undergoing antibiotic treatment (primarily beta-lactams, cephalosporins, and tetracyclines), and cows that had recently calved (within 1–4 days postpartum). Sampling was conducted weekly over a period of 7 weeks, and sample representativeness was ensured by maintaining the milk tank temperature within the range of 3–4 °C and utilizing an automated stirrer to homogenize the milk for 10 min. The homogenized sample was then carefully extracted and transferred to a 500 mL sterile glass bottle. To preserve sample integrity, the bottle was promptly transported to the laboratory.

### 4.3. Experimental Design

An experimental study was carried out to evaluate the effectiveness of the treatment with copper ions on bacteria and antibiotic residues contained in the WM, with longitudinal sampling over seven weeks. The samples were collected, transported to the laboratory, and treated using a time-dependent approach, as follows: (T0) without copper ion treatment, and at 15 and 30 min (T15 and T30) after treatment with copper ions. Chemical and microbiological analyses were then performed on each sample. Samples were collected for each point in time, with three replicates for each analysis.

### 4.4. Copper Ion Treatment

Copper ion treatment was performed in accordance with the protocol described by Steuer et al. (2018) with some modifications [17]. In brief, a glass container containing 500 mL of each WM sample was used, in which two high-purity copper plates were immersed. The plates were stimulated with a low voltage (24 V) and intensity (3 amperes) electrical current to release large concentrations of copper ions. In addition, a magnetic stirrer was inserted into the glass container to homogenize the sample during treatment.

### 4.5. Microbiological Analysis

An analysis was conducted to assess the impact of copper ion treatment on WM. This analysis involved the detection and quantification of bacterial populations relevant to milk quality and animal health. Multiple techniques were employed, including culture with various media and qPCR for mycobacteria. Additionally, a metagenomic analysis technique was used in a single sample to identify the microbial communities before treatment. Details of the different cultivation strategies that were carried out to determine the microbial populations before and after treatment with copper ions are given below:Total Bacterial Count (TBC): 100 µL of 10-fold dilutions of each milk sample were plate duplicated on plate count agar (Oxoid, Hampshire, UK). The plates were incubated at 37 °C for 24 h [19].Total coliforms: 100 µL of 10-fold dilutions of each milk sample were plate duplicated on MacConkey agar (Oxoid, Hampshire, UK), and the plates were incubated at 37 °C for 24 h [19].*Streptococcus*-like colonies: 50 µL of each milk sample were cultured in Edwards medium (Oxoid, Hampshire, UK), and the plates were incubated at 37 °C for 48 h [19].*Staphylococcus* spp. count: 50 µL of each milk sample, were plated on Mannitol salt agar (Oxoid, Hampshire, UK) and the plates were incubated at 37 °C for 48 h [19].Detection of *Listeria* spp.: 100 µL of each milk sample was taken and inoculated into a tube with 5 mL of Listeria Enrichment Broth (Oxoid, Hampshire, UK). The tubes were incubated at 37 °C for 24 h. Then, a 100 µL aliquot of the broth was taken and inoculated into Chromogenic Listeria agar (Oxoid, Hampshire, UK). These plates were incubated at 37 °C for 48 h [38].Detection of *Prototheca* spp.: 100 µL of each milk sample was taken and inoculated into a tube with 5 mL of Prototheca isolation medium (PIM) broth [19]. The tubes were incubated at 37 °C for 24 h. Subsequently, a 100 µL aliquot of the broth was taken and inoculated into PIM agar plates. These plates were incubated at 37 °C for 72 h.Detection of *Salmonella* spp.: 100 µL of each milk sample was taken and inoculated into a tube with 5 mL of Selenite Cysteine Broth (Oxoid, Hampshire, UK). The tubes were incubated at 37 °C for 16 h. Then, a 100 µL aliquot of the incubated broth was taken and inoculated into Xylose Lysine Deoxycholate (XLD) agar plates (Oxoid, Hampshire, UK). These plates were incubated at 37 °C for 48 h [39].Detection of *Mycoplasma* spp.: 100 µL of each milk sample was plated on a modified Hayflick medium (Oxoid, Hampshire, UK) and incubated at 37 °C for 12 days in an atmosphere with 10% CO_2_. The plates were checked after 3, 5, 7, and 12 days for typical colonies of *Mycoplasma* spp. [19].Detection of *E. coli* ESBL and Methicillin-resistant *S. aureus* (MRSA): two methods were employed to detect each pathogen direct plating and enrichment. For both *E. coli* ESBL [40] and MRSA [41], 100 µL of each milk sample was directly plated on CHROMagar^TM^ selective media, ESBL and MRSA, respectively (Chromagar, Paris, France) and incubated at 37 °C for 24 h. Additionally, an enrichment step was performed for each pathogen. Counter samples of the same milk were cultured in Müller–Hinton broth or Müller–Hinton broth + 6.5% NaCl (Oxoid, Hampshire, UK) for 24 h at 37 °C. Subsequently, 50 µL of the enriched broth was plated on the corresponding CHROMagar^TM^ selective medium and incubated under the same conditions.Detection of MAP and *M. bovis* by qPCR: both pathogens were detected using qPCR following a DNA extraction procedure based on phage-mediated separation [42]. However, specific target genes were amplified for each pathogen: IS*900* for MAP [17] and the RD4 gene for *M. bovis* [27].

### 4.6. Metagenomic Analysis

To explore the diversity of bacteria using a non-culture-dependent technique, one of the samples (sample 4) was randomly selected and evaluated before being treated with copper ions using metagenome analysis. An aliquot of 5 mL of milk was taken and centrifuged at 4500× *g* for 10 min, and from the sediment obtained, DNA extraction was carried out using the QIAmp DNA Stool kit (Qiagen, Hilden, Germany), following the instructions of the manufacturer. The concentration and purity of the extracted DNA were evaluated using a NanoDrop ND-1000 spectrophotometer (NanoDrop Technologies, Wilmington, DE, USA). Subsequently, the bacterial DNA was sent to the SeqCenter company (Pittsburgh, PA, USA) to sequence the V3-V4 regions of the *16S rRNA* gene. Samples were prepared using Zymo Research’s Quick-16S kit (Zymo Research, Irvine, CA, USA). Following clean-up and normalization, samples were sequenced on a P1 600cyc NextSeq2000 (Illumina, Inc., San Diego, CA, USA) flow cell to generate 2 × 301 bp PE reads. Quality control and adapter trimming were performed with BCL-convert. Sequences were imported into Qiime2 for subsequent analysis. Primer sequences were removed using Qiime2’s cutadapt plugin. Sequences were denoised using Qiime2’s dada2 plugin and then were assigned operational taxonomic units (OTUs) using the Silva 138 99% OTUs full-length sequence database and the VSEARCH utility within Qiime2’s feature-classifier plugin. OTUs were collapsed to their lowest taxonomic units, and their counts were converted to reflect their relative frequency within a sample.

### 4.7. Detection of Antibiotic Residues

All milk samples were subjected to the detection of antibiotic residues using the IDEXX SNAPduo ST Plus Test rapid test (Idexx Laboratories Inc., Westbrook, ME, USA). This test is based on the detection of beta-lactam antibiotics, cephalosporins, and tetracyclines by binding to enzyme-linked receptors. The reading was carried out according to the manual, and the interpretation considers that if the sample point is lighter than the control point, the sample is positive for the corresponding antibiotic. If the sample point is equal to or darker than the control point, the sample is negative. The sensitivity of the test used for the penicillin group is 4 ppb. For the cephalosporin group, it is 30 ppb, and for the tetracycline group, it is 18 ppb.

### 4.8. Measurement of Copper Concentration in Milk

To determine the total copper concentration in WM samples, all samples were digested with concentrated HCl (37% *w*/*w*) and HNO3 (0.1 N) following the protocol described by Bagherian et al. (2019) [43]. The copper concentration was then measured using an atomic absorption spectrophotometer (AAS) with a GBC SavantAA instrument (GBC Scientific Equipment, Melbourne, Victoria, Australia). Results were reported in mg/L and were based on triplicate measurements.

### 4.9. Data Analysis

To assess data normality, the Shapiro–Wilk test was employed. The Wilcoxon signed-rank test was then used to statistically evaluate any increase or decrease in colony-forming unit (CFU/mL) counts (expressed in log_10_) observed after applying each treatment to the WM samples. All statistical analyses were conducted using the R Studio Program v4.2.1 (R Core Team 2022). A significance level of *p* < 0.05 was chosen to indicate statistically significant differences.

## 5. Conclusions

The present study confirms that treatment with copper ions has a clear deleterious effect on the bacterial populations in WM but showed limited effect on the detection of beta-lactam antibiotics in the samples tested.

## Figures and Tables

**Table 1 antibiotics-13-01085-t001:** Bacterial counts and standard deviation (CFU/mL) in the 7 samples evaluated before copper treatment.

Sample	TBC *	*Streptococcus*	*Staphylococcus*	Coliforms
1	3.6 × 10^5^ ± 1.1 × 10^4^	3.0 × 10^6^ ± 0.0 × 10^0^	9.4 × 10^2^ ± 4.4 × 10^1^	4.5 × 10^4^ ± 2.5 × 10^3^
2	7.7 × 10^4^ ± 9.1 × 10^3^	3.0 × 10^6^ ± 0.0 × 10^0^	1.2 × 10^2^ ± 5.3 × 10^1^	4.4 × 10^3^ ± 2.1 × 10^2^
3	2.4 × 10^7^ ± 9.2 × 10^5^	3.0 × 10^6^ ± 0.0 × 10^0^	1.2 × 10^2^ ± 3.5 × 10^1^	3.0 × 10^4^ ± 2.1 × 10^3^
4	8.4 × 10^5^ ± 2.0 × 10^4^	3.0 × 10^6^ ± 0.0 × 10^0^	2.8 × 10^2^ ± 6.1 × 10^1^	3.5 × 10^4^ ± 1.4 × 10^3^
5	3.9 × 10^4^ ± 8.8 × 10^3^	1.3 × 10^4^ ± 2.3 × 10^3^	1.2 × 10^3^ ± 1.2 × 10^2^	5.1 × 10^2^ ± 6.6 × 10^1^
6	8.5 × 10^4^ ± 3.7 × 10^3^	3.0 × 10^6^ ± 0.0 × 10^0^	6.4 × 10^2^ ± 7.2 × 10^1^	1.6 × 10^3^ ± 2.4 × 10^2^
7	2.4 × 10^6^ ± 1.9 × 10^5^	3.0 × 10^6^ ± 0.0 × 10^0^	4.6 × 10^2^ ± 9.6 × 10^1^	5.6 × 10^2^ ± 6.8 × 10^1^

* Total bacterial counts.

**Table 2 antibiotics-13-01085-t002:** The relative abundance of the most abundant OTUs in sample number 4 was identified through metabarcoding of the *16s RNA* gene.

OTU *	Relative Abundance
*Pseudomonas*	28.45%
*Lactococcus raffinolactis*	14.45%
*Lactococcus lactis*	8.36%
*Escherichia coli*	7.41%
*Lactococcus*	7.13%
*Enterobacteriaceae*	6.91%
*Kluyvera*	6.35%
*Streptococcus uberis*	5.39%
*Chryseobacterium*	3.21%
*Acinetobacter*	2.69%
*Serratia*	1.74%

* Operational Taxonomic Unit.

**Table 3 antibiotics-13-01085-t003:** Average bacteria concentration and standard deviation (CFU/mL) in the 7 waste milk samples before and after 15–30 min treatment with copper ions.

Groups	Minutes of Treatment		
	0	15	30
TBC *			
Sample 1	3.6 × 10^5^ ± 1.1 × 10^4^	8.0 × 10^3^ ± 3.3 × 10^2^	1.0 × 10^2^ ± 4.4 × 10^1^
Sample 2	7.7 × 10^4^ ± 9.1 × 10^3^	2.0 × 10^2^ ± 5.3 × 10^1^	2.0 × 10^1^ ± 1.9 × 10^1^
Sample 3	2.4 × 10^7^ ± 9.2 × 10^5^	1.0 × 10^3^ ± 1.6 × 10^2^	<detection limit
Sample 4	8.4 × 10^5^ ± 2.0 × 10^4^	9.6 × 10^3^ ± 2.0 × 10^2^	6.6 × 10^2^ ± 1.8 × 10^2^
Sample 5	3.9 × 10^4^ ± 8.8 × 10^3^	2.5 × 10^1^ ± 1.5 × 10^1^	1.0 × 10^1^ ± 7.2 × 10^0^
Sample 6	8.5 × 10^4^ ± 3.7 × 10^3^	2.0 × 10^2^ ± 5.3 × 10^1^	<detection limit
Sample 7	2.4 × 10^6^ ± 1.9 × 10^5^	6.0 × 10^1^ ± 3.6 × 10^1^	<detection limit
Average	4.0 × 10^6^	2.7 × 10^3^	1.1 × 10^2^
*Streptococcus*			
Sample 1	3.0 × 10^6^ ± 0.0 × 10^0^	9.6 × 10^2^ ± 2.3 × 10^2^	<detection limit
Sample 2	3.0 × 10^6^ ± 0.0 × 10^0^	6.0 × 10^1^ ± 1.9 × 10^1^	<detection limit
Sample 3	3.0 × 10^6^ ± 0.0 × 10^0^	<detection limit	<detection limit
Sample 4	3.0 × 10^6^ ± 0.0 × 10^0^	<detection limit	<detection limit
Sample 5	1.3 × 10^4^ ± 2.3 × 10^3^	<detection limit	<detection limit
Sample 6	3.0 × 10^6^ ± 0.0 × 10^0^	<detection limit	<detection limit
Sample 7	3.0 × 10^6^ ± 0.0 × 10^0^	<detection limit	<detection limit
Average	2.6 × 10^6^	1.5 × 10^2^	<detection limit
*Staphylococcus*			
Sample 1	9.4 × 10^2^ ± 4.4 × 10^1^	2.0 × 10^1^ ± 1.2 × 10^1^	<detection limit
Sample 2	1.2 × 10^2^ ± 5.3 × 10^1^	2.0 × 10^1^ ± 1.8 × 10^1^	<detection limit
Sample 3	1.2 × 10^2^ ± 3.5 × 10^1^	2.0 × 10^1^ ± 1.8 × 10^1^	<detection limit
Sample 4	2.8 × 10^2^ ± 6.1 × 10^1^	2.0 × 10^1^ ± 1.8 × 10^1^	<detection limit
Sample 5	1.2 × 10^3^ ± 1.2 × 10^2^	<detection limit	<detection limit
Sample 6	6.4 × 10^2^ ± 7.2 × 10^1^	<detection limit	<detection limit
Sample 7	4.6 × 10^2^ ± 9.6 × 10^1^	<detection limit	<detection limit
Average	5.4 × 10^2^	1.1 × 10^1^	<detection limit
Coliforms			
Sample 1	4.5 × 10^4^ ± 2.5 × 10^3^	4.0 × 10^1^ ± 4.0 × 10^1^	<detection limit
Sample 2	4.4 × 10^3^ ± 2.1 × 10^2^	<detection limit	<detection limit
Sample 3	3.0 × 10^4^ ± 2.1 × 10^3^	1.2 × 10^3^ ± 2.0 × 10^2^	<detection limit
Sample 4	3.5 × 10^4^ ± 1.4 × 10^3^	<detection limit	<detection limit
Sample 5	5.1 × 10^2^ ± 6.6 × 10^1^	<detection limit	<detection limit
Sample 6	1.6 × 10^3^ ± 2.4 × 10^2^	<detection limit	<detection limit
Sample 7	5.6 × 10^2^ ± 6.8 × 10^1^	<detection limit	<detection limit
Average	1.7 × 10^4^	1.8 × 10^2^	<detection limit

* Total bacterial counts.

**Table 4 antibiotics-13-01085-t004:** Antibiotic detection results with Idexx SNAPduo ST Plus Test before and after treatment with copper ions.

Sample Name	0	15	30
Sample 1	+	+	+
Sample 2	+	+	+
Sample 3	+	+	+
Sample 4	-	-	-
Sample 5	+	+	-
Sample 6	-	-	-
Sample 7	+	+	Invalid

## Data Availability

All the study data are included in the manuscript submitted.

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
