# Peer review of "Efficacy of Copper Ion Treatment on Bacteria and Antibiotic Residues Contained in Bovine Waste Milk"

_antibiotics, 2024, doi:10.3390/antibiotics13111085_

Round 1
Reviewer 1 Report
Comments and Suggestions for Authors
Comment to authors: antibiotics-3252695
The manuscript entitled " Efficacy of copper ion treatment on bacteria and antibiotic residues contained in bovine waste milk" is reviewed, and the following points are noticed and should be addressed by the authors: Authors have selected novel topic for the study.
1. Can authors provide the recent production data of waste milk? Improved the introduction sections with help of the different study related the effect of copper iron treatment on different types of milk.
2. Line no. 43: check the reference format
3. Liters would be L only and minutes would be min., check throughout manuscript and make uniformity.
4. How effective is copper ion treatment in reducing bacterial load in bovine waste milk?
5. Does the use of copper ions significantly decrease the concentration of antibiotic residues in bovine waste milk?
6. What factors influence the efficacy of copper ion treatment on bacteria and antibiotic residues?
7. How does copper ion treatment compare with other methods of decontamination for waste milk?
8. Are there any potential side effects or risks associated with using copper ions in treating bovine waste milk?
Comments on the Quality of English Language
Improvement is required.
Author Response
Reviewer 1
The manuscript entitled " Efficacy of copper ion treatment on bacteria and antibiotic residues contained in bovine waste milk" is reviewed, and the following points are noticed and should be addressed by the authors: Authors have selected novel topic for the study.
- Can authors provide the recent production data of waste milk? Improved the introduction sections with help of the different study related the effect of copper iron treatment on different types of milk.
We thank the reviewer for the suggestion, nevertheless, waste milk is a neglected topic in which recent information is scarce. Two recent references were added, and a new text was included on lines 47-49
- Line no. 43: check the reference format.
The reviewer is correct, the citation was corrected as suggested on L43.
- Liters would be L only and minutes would be min., check throughout manuscript and make uniformity.
Changes were made as suggested by the reviewer on lines 269-270 for L and 22, 23, 135, 215, 221, 278, and 287 for min.
- How effective is copper ion treatment in reducing bacterial load in bovine waste milk?
We thank the reviewer for the suggestion, the percentage of reduction for each group was added on lines 128-129
- Does the use of copper ions significantly decrease the concentration of antibiotic residues in bovine waste milk?
In this study, we performed only the detection of antibiotic residues with a rapid test to check if there were some differences in the detection before and after the copper treatment. To assess the differences in antibiotic concentration, it is necessary to use quantitative techniques such as HLPC or MS/MS. This limitation is mentioned in lines 238-240. Now, we will start a wider project to investigate the effect of copper treatment on antibiotic concentration and we expect to assess this question.
- What factors influence the efficacy of copper ion treatment on bacteria and antibiotic residues?
Based on the data collected from our experiments, the main factors are the size of the surface of the copper plates, time of exposure, amperage, and volume of milk to treat.
Additionally, the time of treatment, the concentration of antibiotics, and the percentage of fat are described as the main factors that influence the degradation of antibiotics as tetracyclines (Kitazono et al., 2012, 10.1016/j.jhazmat.2012.10.009; Kitazono et al., 2017, 10.1007/s10163-016-0517-9). An explanation was added on lines 238-239.
- How does copper ion treatment compare with other methods of decontamination for waste milk?
The issue inquired is assessed on lines 219-220
- Are there any potential side effects or risks associated with using copper ions in treating bovine waste milk?
We thank the reviewer for the answer. To continue the exploration of this treatment as an alternative to treating waste milk, it is necessary to further investigate the quality of the treated milk in terms of nutritional characteristics and safety. The treatment includes the remotion of the copper to avoid toxicological effects on the calves, but it is necessary to run more tests on the treated milk to ensure that the milk is safe. Additionally, copper can exert selective pressure on bacteria, leading to the emergence of resistant strains mainly through co-selection mechanisms. This issue is mentioned in lines 244-258.
Reviewer 2
The manuscript "Efficacy of copper ion treatment on bacteria and antibiotic residues contained in bovine waste milk" reports the treatment of bacteria and antibiotic residues in bovine waste milk using copper ions. However, the authors should address the following aspects:
- What additional variables could be controlled to ensure that copper ion treatment is more effective in eliminating bacteria and antibiotic residues?
We thank the reviewer for the suggestion. The main variables that can be controlled are the treatment time, which is related to the final copper concentration, and the amperage. This was added on lines 221-223
- How does the effectiveness of copper ions compare with other treatments such as pasteurization or acidification, and under what conditions might copper be the preferable option?
The issue inquired is assessed on lines 219-220
- Is there sufficient evidence to conclude that prolonged use of copper ions will not lead to increased bacterial resistance?
The reviewer is right that bacterial exposure to antimicrobial substances can select resistant bacteria due to selective pressure, as it is discussed on lines 253-258. The rapid and significant reduction in bacterial populations, including potential resistant strains, observed in our study indicates that the treatment exerts a strong antimicrobial effect. Additionally, the high concentrations of copper ions used in our study may further limit the opportunity for bacteria to develop resistance mechanisms.
However, as the reviewer rightly point out, the long-term impact of copper ion treatment on bacterial populations and the potential for resistance development requires further investigation. Our research group is actively exploring this topic.
- How could the evaluation of the long-term effects of copper ion treatment on calf health be improved, especially in terms of growth parameters, feed intake, and nutrient utilization?
The present study is an in-vitro evaluation of the efficacy. To assess the long-term effects of copper ion treatment on calf health, future in vivo studies are essential. Further studies will involve the application of this treatment in milk used for calf feeding and to evaluate the effect on calve’s health. This issue is discussed on lines 244-252.
- Given the high environmental and coliform bacteria count, what improvements could be made to hygiene practices during the handling of milk and feeding equipment to reduce contamination?
We thank the reviewer for the question. Two lines regarding this issue were added on lines 170-172.
- What additional quantitative techniques could be used to more accurately measure the presence of antibiotic residues in milk?
In this study, we performed only the detection of antibiotic residues with a rapid test to check if there were some differences in the detection before and after the copper treatment. To assess the differences in antibiotic concentration, it is necessary to use quantitative techniques such as HLPC or MS/MS. This limitation is mentioned in lines 241-243. Now, we will start a wider project to investigate the effect of copper treatment on antibiotic concentration and we expect to assess this question.
- How could the interaction between copper ions and antibiotic residues be further studied to better understand why limited residue reduction was observed in some analyzed samples?
Currently, we are working to determine the effect in controlled trials. These studies are focused on investigating the effects of copper ions on different classes of antibiotics, examining the influence of milk composition (particularly fat content), on the efficacy of copper ion treatment, and assessing the time effect to study the long-term impact of copper ion treatment on antibiotic residues.
- How can it be ensured that copper ion treatment is equally effective against a broader range of microorganisms, including those that exhibit resistance?
We thank the reviewer for the question. The rapid and significant reduction in bacterial populations, including those with potential resistance mechanisms, observed in our study suggests that copper ion treatment is highly effective against a broad spectrum of microorganisms. Our data demonstrates that less than 0.00001% of microorganisms survived after 30 minutes of treatment. Furthermore, in additional in vitro studies, we have observed 100% efficacy against MRSA, a highly resistant pathogen, and previous research from our group has also shown the effectiveness of copper ion treatment against MAP, a highly resistant bacterium to antibiotics and environmental conditions.
- What additional explanations could be offered for the limited removal of antibiotic residues in some cases?
We thank the reviewer for the question. The limited removal of antibiotic residues may be influenced by several factors, including the initial concentration of the antibiotic, the specific type of antibiotic, and the fat content in the sample. A line was added on line 237.
- How could further investigation into the potential emergence of copper-resistant bacterial strains due to selective pressure be conducted?
While this study represents an initial exploration of copper ion treatment efficacy, it is crucial to acknowledge the potential for the development of copper-resistant bacteria through selective pressure. To address this concern, future studies could incorporate long-term exposure and whole genome sequencing and gene expression studies before and after copper ion treatment.
- What practical measures could be taken to reduce environmental and fecal contamination in milk samples?
We thank the reviewer for the question. Two lines regarding this issue were added on lines 170-172.
- How could metagenomic analysis be expanded to provide an even more detailed view of the microbial communities present in waste milk?
To gain a deeper understanding of the microbial communities in waste milk, could be implemented new studies with bigger sample sizes and longitudinal studies and incorporate milk from different farms.
- Given that Coxiella burnetii was found in the samples, what additional monitoring measures could be implemented on farms to control the spread of this pathogen?
We thank the reviewer for the comment. While the primary focus of this study was not on Coxiella burnetii, a line was added to complement the discussion on lines 206-207. Our previous investigation of the prevalence based on bulk tank milk analysis shows this microorganism is present in a high percentage of farms, but only a few cases of sick animals or humans have been described. It seems that in the conditions of dairy milk production in Chile, not only the presence of these pathogens is required, but a high density of animals in the calving season is needed to trigger the outbreaks.
- What other types of samples could offer greater sensitivity for detecting M. bovis and MAP in future studies?
This question was assessed in lines 179-181. “While milk samples can be used for such purposes, fecal or blood samples generally offer greater specificity and sensitivity for these pathogens”.
- What additional strategies could be implemented to prevent the transmission of S. aureus to calves through contaminated waste milk?
We thank the reviewer for the comment. This issue was added on lines 200-201.
- Since antibiotic residues were detected in 70% of the samples, what actions could be taken to improve the control of antibiotic use in dairy cattle?
We thank the reviewer for this comment. While the control of antibiotic use in dairy cattle is an important topic, a deeper discussion on this subject is beyond the scope of the present study. However, we acknowledge that the detection of antibiotic residues in 70% of WM samples aligns with existing findings, as WM commonly contains antibiotic residues. Addressing this issue would indeed require stricter monitoring protocols and judicious antibiotic use practices, which could be valuable areas for further research.
additional hypotheses could be tested to better understand the interaction between copper ions and antibiotic residues?
We thank the reviewer for this suggestion. Future studies could explore additional hypotheses regarding the interactions between copper ions and various types of antibiotics to deepen our understanding of these dynamics. For instance, investigating how different classes of antibiotics interact with copper ions, as well as examining concentration-dependent effects, could provide more insight into the underlying mechanisms and potential implications for the possible practical use of copper treatment of WM.

Reviewer 2 Report
Comments and Suggestions for Authors
The manuscript "Efficacy of copper ion treatment on bacteria and antibiotic residues contained in bovine waste milk" reports the treatment of bacteria and antibiotic residues in bovine waste milk using copper ions. However, the authors should address the following aspects:
· What additional variables could be controlled to ensure that copper ion treatment is more effective in eliminating bacteria and antibiotic residues?
· How does the effectiveness of copper ions compare with other treatments such as pasteurization or acidification, and under what conditions might copper be the preferable option?
· Is there sufficient evidence to conclude that prolonged use of copper ions will not lead to increased bacterial resistance?
· How could the evaluation of the long-term effects of copper ion treatment on calf health be improved, especially in terms of growth parameters, feed intake, and nutrient utilization?
· Given the high environmental and coliform bacteria count, what improvements could be made to hygiene practices during the handling of milk and feeding equipment to reduce contamination?
· What additional quantitative techniques could be used to more accurately measure the presence of antibiotic residues in milk?
· How could the interaction between copper ions and antibiotic residues be further studied to better understand why limited residue reduction was observed in some analyzed samples?
· How can it be ensured that copper ion treatment is equally effective against a broader range of microorganisms, including those that exhibit resistance?
· What additional explanations could be offered for the limited removal of antibiotic residues in some cases?
· How could further investigation into the potential emergence of copper-resistant bacterial strains due to selective pressure be conducted?
· What practical measures could be taken to reduce environmental and fecal contamination in milk samples?
· How could metagenomic analysis be expanded to provide an even more detailed view of the microbial communities present in waste milk?
· Given that Coxiella burnetii was found in the samples, what additional monitoring measures could be implemented on farms to control the spread of this pathogen?
· What other types of samples could offer greater sensitivity for detecting M. bovis and MAP in future studies?
· What additional strategies could be implemented to prevent the transmission of S. aureus to calves through contaminated waste milk?
· Since antibiotic residues were detected in 70% of the samples, what actions could be taken to improve the control of antibiotic use in dairy cattle?
· What additional hypotheses could be tested to better understand the interaction between copper ions and antibiotic residues?
Author Response

(The authors gave the same response as above.)

Round 2
Reviewer 1 Report
Comments and Suggestions for Authors
Authors have improved the manuscript and can recommended for publication.
Reviewer 2 Report
Comments and Suggestions for Authors
The authors of the manuscript “Efficacy of Copper Ion Treatment on Bacteria and Antibiotic Residues Contained in Bovine Waste Milk” have made the necessary corrections suggested by the reviewers. The manuscript shows a notable improvement and can be published in its current form on the journal's platform.